# Dissemination of information in event-based surveillance, a case study of Avian Influenza

**Sarah Valentin**[1,2,3,4☯], **Bahdja Boudoua**[1,2☯], **Kara Sewalk**[5], **Nejat Arınık**[2], **Mathieu Roche**[1,2,3], **Renaud Lancelot**[1,3], **Elena Arsevska**[1,3]*

1 Joint Research Unit Animal, Health, Territories, Risks, Ecosystems (UMR ASTRE), French Agricultural Research Centre for International Development (CIRAD), National Research Institute for Agriculture, Food and Environment (INRAE), Montpellier, France, 2 Joint Research Unit Land, Environment, Remote Sensing and Spatial Information (UMR TETIS), Université de Montpellier, AgroParisTech, French Agricultural Research Centre for International Development (CIRAD), French National Centre for Scientific Research (CNRS), National Research Institute for Agriculture, Food and Environment (INRAE), Montpellier, France, 3 French Agricultural Research Centre for International Development (CIRAD), Montpellier, France, 4 Département de biologie, Université de Sherbrooke, Sherbrooke, Canada, 5 Computational Epidemiology Group, Boston Children's Hospital, Boston, MA, United States of America

☯ These authors contributed equally to this work.
* elena.arsevska@cirad.fr

**Data Availability Statement:** Data used in this paper are available from the Zenado database at doi (https://doi.org/10.5281/zenodo.6908000). The

## Abstract

Event-Based Surveillance (EBS) tools, such as HealthMap and PADI-web, monitor online news reports and other unofficial sources, with the primary aim to provide timely information to users from health agencies on disease outbreaks occurring worldwide. In this work, we describe how outbreak-related information disseminates from a primary source, via a secondary source, to a definitive aggregator, an EBS tool, during the 2018/19 avian influenza season. We analysed 337 news items from the PADI-web and 115 news articles from HealthMap EBS tools reporting avian influenza outbreaks in birds worldwide between July 2018 and June 2019. We used the sources cited in the news to trace the path of each outbreak. We built a directed network with nodes representing the sources (characterised by type, specialisation, and geographical focus) and edges representing the flow of information. We calculated the degree as a centrality measure to determine the importance of the nodes in information dissemination. We analysed the role of the sources in early detection (detection of an event before its official notification) to the World Organisation for Animal Health (WOAH) and late detection. A total of 23% and 43% of the avian influenza outbreaks detected by the PADI-web and HealthMap, respectively, were shared on time before their notification. For both tools, national and local veterinary authorities were the primary sources of early detection. The early detection component mainly relied on the dissemination of nationally acknowledged events by online news and press agencies, bypassing international reporting to the WAOH. WOAH was the major secondary source for late detection, occupying a central position between national authorities and disseminator sources, such as online news. PADI-web and HealthMap were highly complementary in terms of detected sources, explaining why 90% of the events were detected by only one of the tools. We show that current EBS tools can provide timely outbreak-related information and priority news sources to improve digital disease surveillance.

script for our results presented in the paper are available in a public GitHub repository (https://github.com/SarahVal/EBS-network).

**Funding:** This project has received funding from the European Union's Horizon 2020 research and innovation programme under grant agreement No 874850 and is catalogued as MOOD 049.

## Introduction

Recent developments in internet and digital technologies have contributed to the establishment of the Epidemic Intelligence (EI) framework, aiming at the early identification of potential health threats from sources of intelligence of any nature, their verification, and assessment for timely prevention and control by public and animal health (PH/AH) agencies. Event-based surveillance (EBS), as part of the EI, gathers unstructured data on potential and non-verified disease outbreaks mainly by monitoring the web, such as online media, social networks, and blogs. The EBS is complementary to traditional, indicator-based surveillance (IBS), also part of the EI, which collects structured data on verified disease outbreaks through routine national surveillance systems [1–3].

Since the early 2000s, several automatised EBS tools with open-access have been created, such as HealthMap, operating since 2006 and monitoring web sources for the public, animal, and plant health threats [4], and PADI-web, operating since 2016 and monitoring web sources for mainly animal health threats [5]. The two open-access tools are used for the detection and monitoring of potential outbreaks reported in non-official sources on the web, including known diseases, such as avian influenza or Ebola [6, 7], or clinical signs of unknown origin, such as acute respiratory syndrome [8]. The main users of the two tools are EI staff at national and supranational PH/AH agencies and organizations, among others such as the French Platform for epidemiological surveillance in animal health (Platform ESA) [7] and the European Centre for Disease Control (ECDC) [9].

Both HealthMap and PADI-web implement algorithms to capture news on potential disease outbreaks from a broad range of data sources on the web in multiple languages and geographical regions [4, 5]. For example, HealthMap gathers data from Baidu, SoSo, Google News aggregators, and ProMED-mail in nine languages. PADI-web collects data from the Google News aggregator in 16 languages. Both tools further implement classification and information extraction algorithms to filter and extract the relevant outbreak information in a structured format from the free text, such as the place, date, and host of a described outbreak. Finally, HealthMap provides users with a world map interface to visualise the reports and information sources that report outbreaks. PADI-web provides users with a list of information sources and news content that reports outbreaks.

Previous evaluations of the EBS tools in use today, including HealthMap and PADI-web, focused mainly on the assessment of their extrinsic performance, such as timeliness, positive predictive value, or sensitivity (Se) in detecting outbreaks from the sources they monitor, compared to official disease outbreaks [6, 7]. From an end-user perspective, Barboza et al. [10, 11] assessed metrics such as the usefulness, simplicity, and flexibility of an EBS tool.

The understanding of the role of the inputs (i.e. the monitored sources) on the performance of EBS tools is less explored. Barboza et al. [10] found that the type of moderation, sources, languages, regions of occurrence, and types of cases influence EBS tool performance. Schwind et al. [12] identified that domestic and national news sources were more likely to report outbreaks than international news portals.

This study aimed to fill the existing gap in the role of sources monitored by EBS tools. We consider EBS tools as aggregators which collect disease outbreak information at the end of a transmission chain, referred to as a network. More precisely, we aimed to characterise the sources of outbreak information detected by an EBS tool and assess how the sanitary information circulates through the monitored sources before being detected by an EBS tool.

We assessed the flow of outbreak information from primary sources, providers of the information, until the end sources, EBS tools, and final aggregators of the information. We represent this information flow through a network structure. Moreover, we provide an in-depth

analysis of the extracted networks and the characteristics of the sources involved in outbreak reporting using two EBS tools, HealthMap and PADI-web. In this study, we address three main questions:

1. What are the sources involved in the reporting of outbreak-related information on the web?

2. What are the roles of the different sources regarding the dissemination of outbreak-related information on the web, and what are their characteristics in terms of type, specialisation, and geographical scope?

3. How complementary are the different EBS tools in terms of monitored sources and reported outbreak-related information?

In this study, we further propose a new representation of the sources and their networks involved in digital disease surveillance to improve the detection and analysis of signals of disease emergence from online media. This representation and associated analysis address these questions.

The remainder of this paper is organised as follows. First, we summarise the objectives and methods of assessing information dissemination across data (news) sources. Next, we detail our methodology to collect and assess the dissemination of outbreak-related information via PADI-web and HealthMap. We present and discuss our results in Section 3, before summarising the main conclusions of our work.

## Materials and methods

### Data collection

To conduct this study, we chose to analyse news reports of Avian Influenza (AI) detected by two EBS tools, PADI-web and HealthMap. AI viruses can spread over long distances via trade in poultry and wild-caught birds, as well as via the movement of wild birds [13]. AI outbreaks are responsible for significant economic losses resulting from trade restrictions, loss of disease-free status for affected countries, or culling measures in infected flocks. Moreover, AI has great zoonotic potential, as some subtypes can infect different avian and mammalian animal hosts, including humans [14]. Thus, early detection of AI outbreaks is essential for implementing protection and control measures and helping contain their spread.

For our study, we extracted all English news reports from PADI-web and HealthMap EBS tools, which described one or several AI outbreaks and were published between 1 July 2018 and 30 June 2019 (i.e. 337 news reports from PADI-web and 115 news reports from HealthMap). We chose a one-year study period (July 2018 to June 2019) to capture the spatiotemporal epidemiological characteristics of AI outbreaks worldwide. The detection of the virus at a specific date and time is hereafter referred to as an event (most events are outbreaks, but some describe the detection of the virus in the environment). Two epidemiologists (BB, SV, authors of this work) manually assessed the relevance of each news item (a report was considered relevant if it contained at least one event) and discarded irrelevant news. Importantly, the events can be either reported as confirmed or suspected, as one of the keystones of EI is the detection of potential outbreaks before official confirmation.

### Event detection

Two epidemiologists (BB and SV, authors of this work) read the relevant news and identified all reported events. Each event described in the detected news was classified as official or non-official.

Official events corresponded to outbreaks officially notified by AH authorities. For this purpose, we used the Emergency Prevention System for Priority Animal and Plant Pests and Diseases (EMPRES-i), a global animal health information system [15, 16] developed by the Food and Agriculture Organization (FAO) of the United Nations. EMPRES-i allows free access to and sharing of disease outbreak data to support data analysis and notification to national AH authorities by monitoring and summarising the global status of priority animal diseases and zoonoses, including AI. One of the main sources of information for the EMPRES-i is the verified disease outbreak data provided by national AH authorities, mainly through traditional disease surveillance by the World Organisation for Animal Health (WOAH). The EMPRES-i has tracked AI outbreaks since 2003.

When an event could not be linked to an official event from the EMPRES-i, we labelled it as non-official and recorded the epidemiological information provided in the report (i.e. subtype, reported date of the event, the country and location of the event, the host affected, and the number of cases). This enabled us to identify when the same non-official event was reported in different news articles.

For both official and non-official events, we calculated the number of non-overlapping events between the two EBS tools, that is, the events that were detected by one tool out of two.

For the official events, we evaluated the sensitivity (Se) and timeliness of each tool. Timeliness is the lag in days between the date of official notification to the WOAH (day 0), as recorded in the EMPRES-i database, and the date when the same event was first detected by the PADI-web and HealthMap. A negative lag means that the EBS tool detects an event in a timely manner, that is, before the date of notification. A positive lag indicated that the EBS tool was untimely for detecting an outbreak, that is, the same day or after the official notification date. Se is defined as the ability of the EBS tool to report an event present in the EMPRES-i database, corresponding to the proportion of true positive events (TP) among the sum of true positive and false-negative (FN) events (Se = TP/(TP+FN)). A TP event was defined as all AI outbreaks in the EMPRES-i database during the study period. An FN event was defined as an event present in the EMPRES-i database that was not detected by an EBS tool. The specificity of event-based surveillance tools cannot be calculated, as it is impossible to assess the status of non-official events detected [11]; there may be false positive events as well as TP events not reported to the gold standard databases (WOAH and EMPRES-i).

## Network construction

To trace back the primary sources, we manually traced the information pathways of all events mentioned in the PADI-web and HealthMap news. We assumed that an information pathway could be deduced from the sources cited in the news content. In the information pathway, the first node is called the primary source (i.e. the earliest emitter source), the last node is called the final source (i.e. the final aggregator, PADI-web, or HealthMap), and the remaining nodes, if any, are called secondary sources. The combination of all information pathways from news events gives a network structure, referred to as a network of information pathways.

Let $G = (V, E, A)$ be a directed unweighted attributed graph representing a network of information pathways, where V, E, and A are the set of network nodes, network edges, and attributes associated with the nodes, respectively [17]. The network nodes represent the sources and final aggregators (PADI-web and HealthMap). Each node has three attributes, as defined in S1 Table: type (e.g. online news source, national veterinary authority, etc.), geographical focus (local, national, or international), and specialisation in animal health news coverage (general or specialised). The edges represent the dissemination of event information between two nodes (an emitter source, $S_E$ that sends the event, and a receptor source, $S_R$ that receives

the event). The graph is directed as the information is transmitted from the $S_E$ to the $S_R$. A directed graph is formally defined as a graph G for which each edge in $E$ has an ordering to its vertices (i.e. such that $e_1 = (u,v)$ is distinct from $e_2 = (v,u)$, for $e_1,e_2 \in E$). In our approach, the edges are not weighed because we create an edge between an $S_E$ and $S_R$ if $S_R$ cites $S_E$ at least once.

It is worth noting that an event can be transmitted through several paths and that a path can transmit several events. The first case occurs when the same event is reported by different sources (e.g. two online news articles). The second occurs when a single news article reports several events. Based on this fact, we separated the global graph into three subgraphs depending on the type of events detected and their timeliness: a graph containing the paths associated with the early detection of official events (timeliness $< 0$), a graph containing the paths associated with the late detection of official events (timeliness $\geq 0$), and a graph containing the paths associated with the detection of non-official events.

## Network analysis

**Network description.** We first describe the network of information pathways extracted from the PADI-web and HealthMap news, PADI-web, and HealthMap networks hereafter, in terms of the number of edges, nodes, and paths. We visualised the networks using a chord diagram and classified the nodes according to their source types.

**Path analysis.** To evaluate the network performance regarding the dissemination of health events, we calculated the path length and reactivity of the networks. The path length is the number of edges in the path. The path length corresponds to the number of secondary sources between the primary and final aggregators (PADI-web or HealthMap); for example, a path composed of three edges contain two secondary sources. We hypothesised that the fewer the number of sources in a path, the faster the transmission of information.

Path reactivity is the sum of the time lags between all the nodes composing the path. Path reactivity measures the number of days between the primary source's communication and detection by the final aggregator. Path reactivity is highly relevant for EI because it reflects the ability of the system to quickly disseminate events to the aggregator.

**Node analysis.** We assessed the importance of the nodes, i.e., the sources, in the PADI-web and HealthMap networks using qualitative and quantitative attributes.

We first evaluated the global ability of the sources to receive and transmit event information by merging PADI-web and HealthMap networks. We calculated the in-degree, out-degree, and all-degree centrality measures of nodes [18] and analysed their distribution according to the type of source. In-degree is the number of incoming edges to a node; thus, sources with a high in-degree collect information from a large range of other sources. Out-degree is the number of outcoming edges from a node. Sources with a high out-degree are often cited; thus, they can communicate outbreak-related information with high visibility. The all-degree is the sum of the in-degree and out-degree. Sources with a high all-degree, also referred to as "hubs", combine the capacity to receive and share outbreak-related information [19].

We further analysed the role of the sources in the different subgraphs (early, late, and non-official), separating the PADI web and HealthMap networks. We classified the sources according to their location in the network (primary versus secondary) and calculated the frequency of each type of source (e.g. online news). We further calculated the proportion of primary and secondary sources according to their geographical focus and specialisation.

**Software.** The database was constructed using MS Office Access (version 2019). The analysis was performed using the *igraph* package available in R version 3.6 [20].

## Results

### Event detection

Between 1 July 2018 and 30 June 2019 national animal health authorities reported 351 AI outbreaks in the WOAH. Among these, 81% (284/351) were from domestic birds, 10% (34/351) were from wild birds, 6% (24/351) were from environmental samples, and 3% (12/351) were unspecified.

The PADI-web detected 408 unique AI outbreak-related news reports, 337 (83%) of which were considered relevant after manual curation (see details in S2 Table). HealthMap detected 163 unique AI outbreak-related news reports, 115 (71%) of which were relevant after manual curation. Among the relevant reports, 37 were detected using both the EBS systems.

Both the PADI-web and HealthMap had a median of one event per news report (min = 1, max = 14). In the PADI-web relevant news reports, 230 events were described, including 193 events that were not detected by HealthMap (Table 1). Among the detected events, 87% (199/230) were official events; that is, they matched a notified AI outbreak to the WOAH. The remaining 31 events (13%) were unofficial, that is, they could not be verified. The majority (82%) of PADI-web events described AI outbreaks in domestic birds (185/226), while AI outbreaks in wild birds represented 13% (29/226) of the events.

HealthMap relevant reports described 68 events, among which 31 did not overlap with PADI-web detected events (Table 1). Among these events, 88% (60/68) were official and 12% (8/68) were non-official. Similar to the PADI-web, 78% (53/68) of the HealthMap events were in domestic birds, whereas 16% (11/68) were in wild birds.

The non-overlapping events represented 45% (222/489) of all events detected by PADI-web and HealthMap. The number of events reported to the WOAH and the events detected by the two EBS tools per week and region are provided in the S3 Table.

The Se of HealthMap and PADI-web were 17% (60/351) and 57% (199/351), respectively. Based on a McNemar statistical test, the sensitivities differ significantly at the standard significance level of 5% (see S1 File).

The timeliness of PADI-web varied from 112 days before to 39 days after notification of an outbreak to the WOAH; 24% (47/199) of the events detected by PADI-web were detected before their official notification, representing 13% of the official events (Fig 1). The PADI-web was timelier in detecting AI events in wild birds than in domestic birds. More precisely, 21% (36/174) of the AI outbreaks in domestic birds in the PADI-web were detected before their official notification, while 56% of the events (9/16) were detected early in wild birds, with a maximum of 112 days before official notification in wild birds.

The timeliness of HealthMap varied from 46 days before to 66 days after an official reporting of an event to the WOAH; 43% (26/60) of the events detected by the tool were reported

**Table 1. Number of official and non-official events of AI detected by PADI-web and HealthMap between July 2018 and June 2019.** The number of non-overlapping events is shown between parentheses.

| Type of host | PADI-web | | HealthMap | |
| --- | --- | --- | --- | --- |
|  | Official | Non-official | Official | Non-official |
| Domestic birds | 174 (147) | 15 (13) | 48 (23) | 5 (3) |
| Wild birds | 16 (10) | 13 (12) | 9 (3) | 2 (1) |
| Mammals | - | 2 (1) | - | 1 (0) |
| Environmental | 8 (8) | - | 2 (0) | - |
| Unspecified | 1 (1) | 1 (1) | 1 (1) | - |
| Total | 199 (166) | 31 (27) | 60 (27) | 8 (4) |

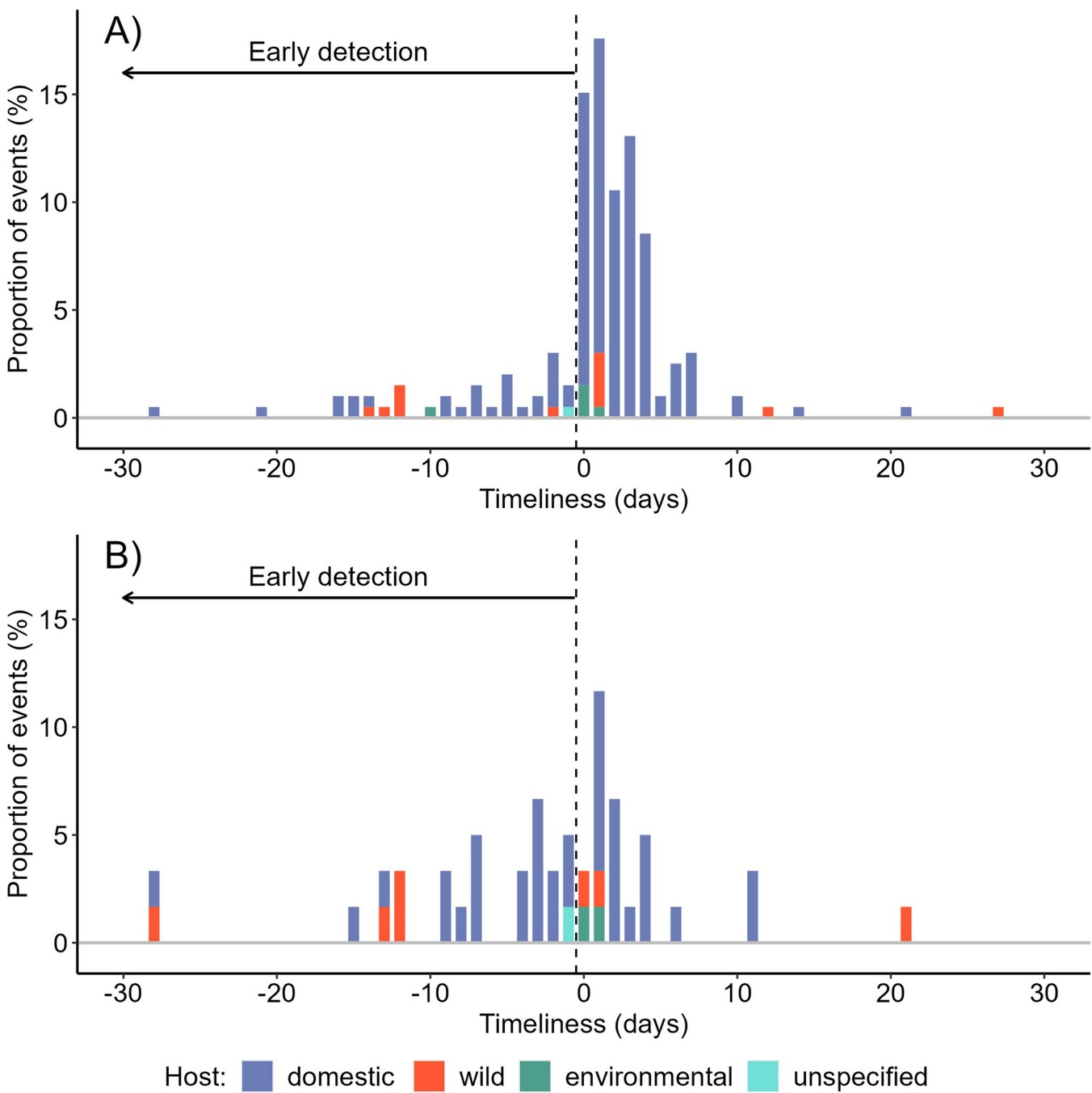

**Fig 1.** Timeliness in the detection of AI outbreaks according to the type of host for A) PADI-web and B) HealthMap. For visibility, extreme values i.e., less than 30 days and higher than 30 days are not shown.

before the official notification, representing 7% of the official events (Fig 1). In the HealthMap network, 42% (20/48) and 56% (5/9) of AI outbreaks in domestic and wild birds, respectively, were detected before their official notification, with a maximum of 43 days before official notification in wild birds.

## Network analysis

**Network description.** During the study period, the PADI-web network disseminated AI outbreak-related information from 250 different nodes (sources), 446 unique edges (links), and 455 paths. The HealthMap network comprised 108 nodes, 150 unique edges, and 107 paths. A graphical representation of both networks, as well as details of the edges and nodes, are provided in S4–S7 Tables and S1 Fig.

Online news was the most represented source (47.6% of the sources in the PADI-web network and 36% in the HealthMap network (Table 2). Local veterinary authorities were more frequent in the PADI web network than in the HealthMap network. Conversely, press agencies represented 10.2% of the HealthMap network sources, compared to 4.8% in the PADI-web network.

**Path analysis.** Most of the PADI-web paths are composed of two (232/455; 51%) and three (182/455; 40%) edges, 4% (18/455) of the paths are composed of a single edge (they do not cite any source), and 5% (21/455) of the paths are made up of four edges and more. Similarly, most HealthMap paths are composed of two (53/107; 50%) and three (32/107; 30%) edges, 14% (15/107) of the paths are composed of one edge and 5% (7/107) are composed of five edges.

In the PADI-web, 83% (376/455) of the paths propagated events in one day (n = 41) or less than one day (n = 335). Similar results were observed in HealthMap, with 94% (87/107) of the paths propagating events in one day (n = 3) or less than one day (n = 84).

**Quantitative node analysis.** Only 24% (69/287) of the sources in the global network of the PADI-web and HealthMap were characterised by an in-degree greater than 1, indicating that most of the sources received information from a single source. The EBS tools, PADI-web and HealthMap, international veterinary authority, social platforms, press agencies, and research organisations had the highest median in-degrees (Fig 2).

These groups contain sources which have access to a large amount of information, that is, different sources. The EBS tools had the highest median in-degree because they included PADI-web and HealthMap, the two aggregators in our study. Except for these two EBS tools, the WOAH stood out with a maximal in-degree equal to 25. Online news sources were characterised by a median in-degree of one, but twelve outliers had an in-degree higher than 5,

**Table 2. Types of sources (i.e., nodes) in PADI-web and HealthMap networks disseminating outbreak-related news on Avian influenza between 1 July 2018 and 30 June 2019.**

| Type of source | PADI-web | HealthMap |
|---|---|---|
| online news source | 47.6% (n = 119) | 36.1% (n = 39) |
| national vet authority | 14% (n = 35) | 20.4% (n = 22) |
| local veterinary authority | 13.2% (n = 33) | 8.3% (n = 9) |
| local official authority | 6% (n = 15) | 3.7% (n = 4) |
| press agency | 4.8% (n = 12) | 10.2% (n = 11) |
| radio, TV | 4.4% (n = 11) | 3.7% (n = 4) |
| laboratory | 2.4% (n = 6) | 2.8% (n = 3) |
| national official authority | 2% (n = 5) | 5.6% (n = 6) |
| research organisation | 1.6% (n = 4) | 1.9% (n = 2) |
| local person | 1.2% (n = 3) | 0 |
| social platform | 1.2% (n = 3) | 4.6% (n = 5) |
| private company | 0.8% (n = 2) | 0 |
| EBS tool | 0.4% (n = 1) | 1.9% (n = 2) |
| international veterinary authority | 0.4% (n = 1) | 0.9% (n = 1) |
| Total | 250 | 108 |

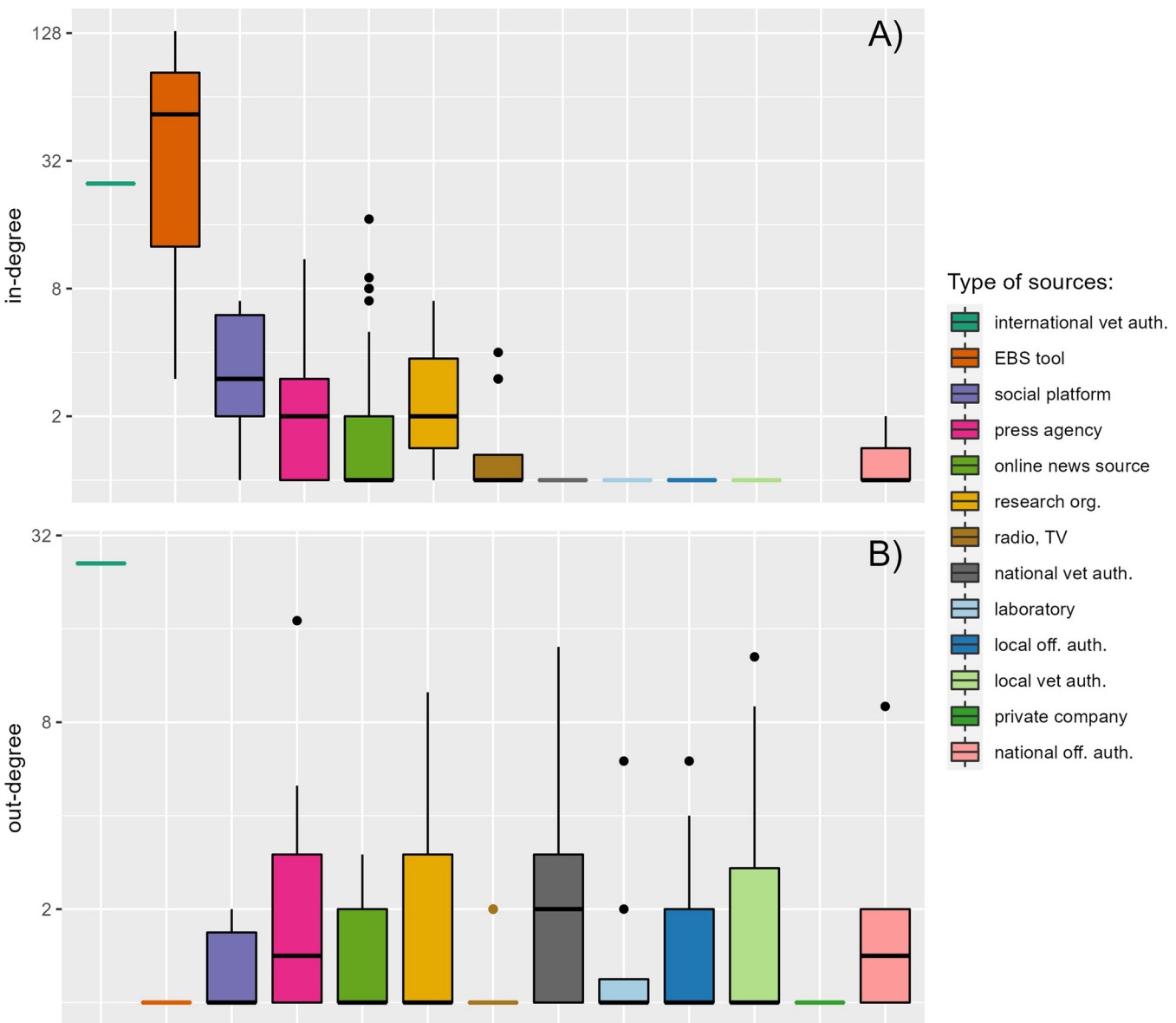

**Fig 2.** Performance of sources in terms of A) in-degree and B) out-degree, aggregated by type. The y-axis has been log-scaled. Distributions of in-degree and out-degree are represented with box plots based on a 95% confidence interval (outliers are represented with dots).

among which "Times of India", and two sources specialised in poultry production, "Poultry-Site" and "WATTAgNet" (Table 3). Similarly, the social platforms, press agencies, and research organisations were characterised by a high intra-group variance, containing highly connected sources (e.g. Reuters, Xinhua).

The median out-degree of nine out of the 13 types of sources was one, explained by the fact that 64% (183/297) of the sources in the networks were cited only once. Local and national veterinary authorities had higher out-degree values than in-degree values, highlighting their role as sources of information. Individually, the WOAH stands out with the maximal out-degree of 26, followed by Reuters, one national authority, and one local veterinary authority (Table 3). As for in-degree, the out-degree variance was high in most groups, owing to the presence of outliers being significantly better transmitters than the other sources of their group.

**Table 3. Top-5 sources in terms of in-degree, out-degree and all-degree.** The EBS tools PADI-web and HealthMap were excluded as they were chosen as the aggregators in our study.

| | Source | Value | Type |
|---|---|---|---|
| **In-degree** | WOAH | 25 | International vet. auth. |
| | Times of India | 17 | Online news |
| | Xinhua | 11 | Press agency |
| | The Poultry Site | 9 | Online news |
| | WATTAgNet | 8 | Online news |
| **Out-degree** | WOAH | 26 | International vet. auth. |
| | Reuters | 17 | Press agency |
| | Bulgaria Vet Auth | 14 | National vet. auth. |
| | Minnesota Vet Authorities | 13 | Local vet. auth. |
| | USA National Oceanic and Atmospheric Administration | 10 | Research org. |
| **All-degree** | WOAH | 51 | International vet. auth. |
| | Reuters | 24 | Press agency |
| | Times of India | 20 | Online news |
| | Bulgaria Vet Auth | 15 | National vet. auth. |
| | Xinhua | 14 | Press agency |

WOAH was the best-performing source in terms of all degrees, confirming its central position. It was followed by two press agencies, Reuters and Xinhua, the veterinary authority of Bulgaria, and Indian online news, Time of India (Table 3).

**Qualitative nodes analysis.** National veterinary authorities were the most frequent primary source of events in the late detection of events in both HealthMap and PADI-web (69% and 63% of the primary sources, respectively) and the early detection of HealthMap events (42% of the secondary sources) (Figs 3 and 4; detailed numbers in S8 and S9 Tables). Local veterinary authorities were the most frequent primary source involved in the early detection of events by the PADI-web (44% of the primary sources) and the second most frequent in

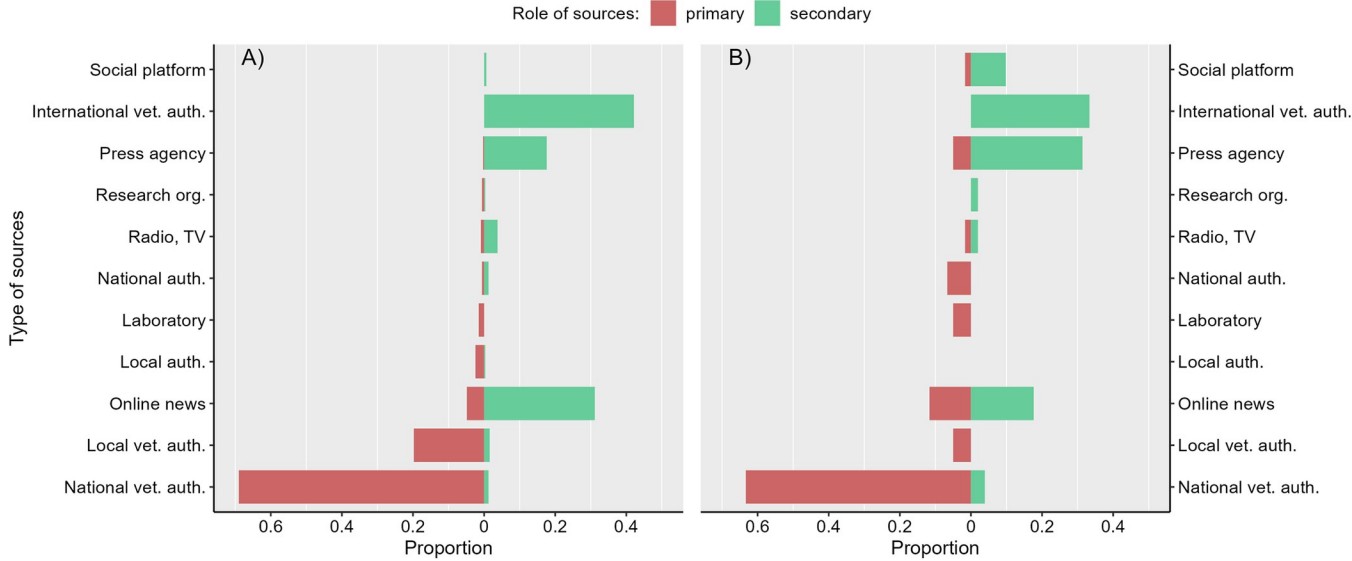

**Fig 3.** Proportion of the types of primary and secondary sources according to their role in the (a) PADI-web and (b) HealthMap late detection networks. Primary sources are sources that are the first to emit an event, secondary sources are sources which receive and emit an event to another source.

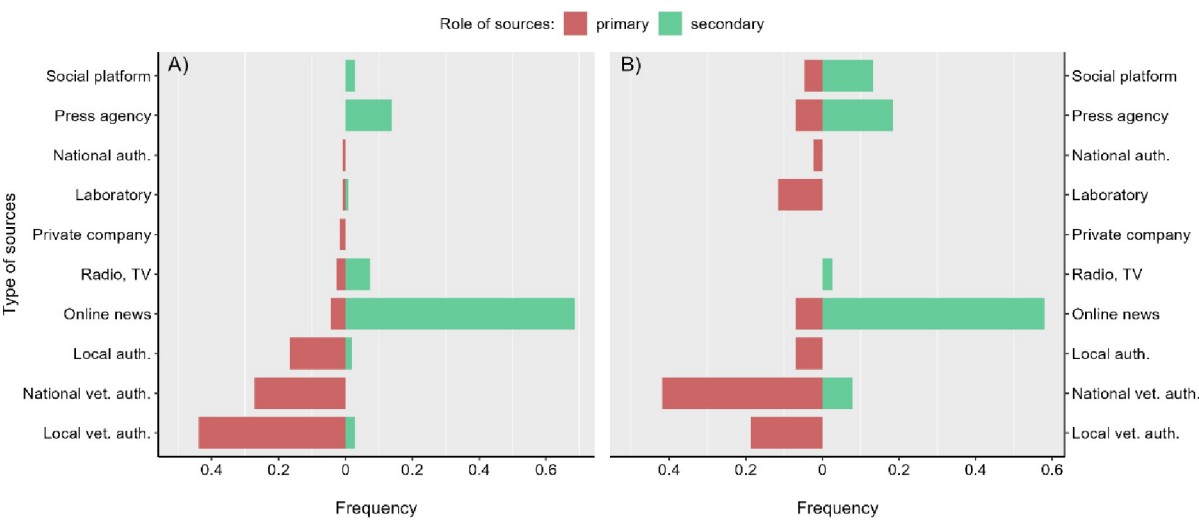

**Fig 4.** Proportion of the types of primary and secondary sources according to their role in the (a) PADI-web and (b) HealthMap early detection network. Primary sources are sources that are the first to emit an event, secondary sources are sources which receive and emit an event to another source.

HealthMap. The transmission of events in the late detection context was mainly driven by WOAH, press agencies, and online news for both the EBS tools. The transmission of events in the early detection context was mainly driven by online news sources (69% and 58% of the secondary sources in PADI-web and HealthMap, respectively), and press agencies were less frequent than in the early detection networks.

Social platforms represented 13% of the secondary sources involved in the early detection by HealthMap, whereas this type of source was barely used by the PADI-web.

Nearly 75% of the primary sources in the early detection network of the PADI-web had a local geographical scope, in contrast to 26% in HealthMap (Fig 5). This result was consistent with our previous results, highlighting the role of local sources in the early warning of disease outbreaks. The late detection networks mainly relied on sources with a national scope for both EBS tools, corresponding to the role of the national veterinary authorities.

Early detection networks relied on both national and international sources as intermediates, while late detection was mostly driven by international sources, as explained by the role of the WOAH in the official communication of events in the news.

Specialisation showed the same pattern between late and early detection and between the EBS tools, with at least 75% of the primary sources being specialised (S1 Fig).

## Discussion

In this work, we described how outbreak-related information circulates in news sources captured by two EBS tools, PADI-web and HealthMap. We assessed the EBS tools network, including primary and secondary sources, and their characteristics in terms of type, geographical scope, specialisation, and importance in the dissemination of information using network centrality metrics. In addition, we assessed the timeliness of sharing officially notified AI outbreak information.

### Global performances of PADI-web and HealthMap networks

PADI-web and HealthMap, to varying extents, capture false positive news reports (with respective report precisions of 83% and 71%, respectively). Even if considered irrelevant for

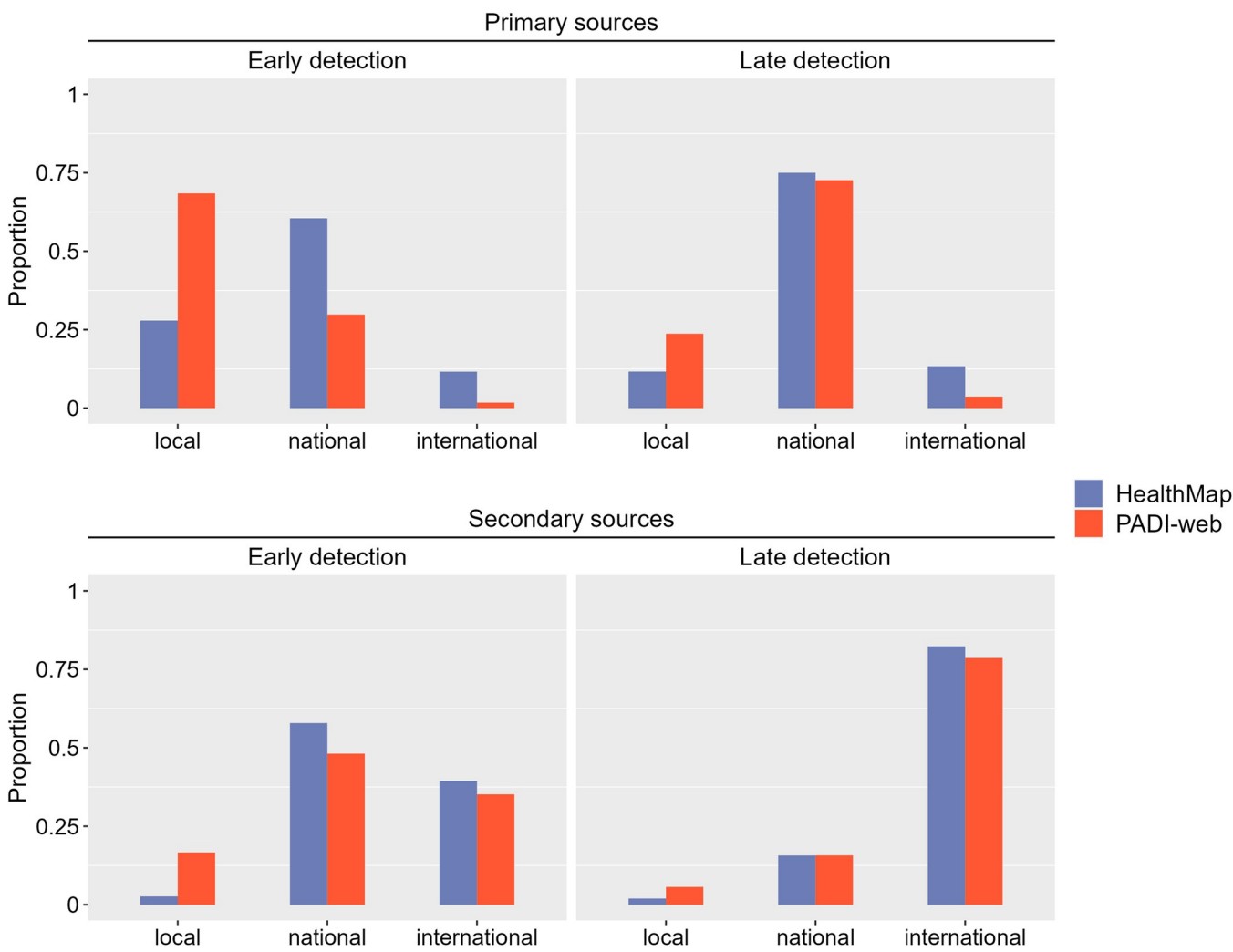

**Fig 5. Proportion of the geographic scope of primary and secondary sources in the PADI-web and HealthMap early and late detection networks.**

this study, most discarded news reports were related to AI events and contained contextual epidemiological information useful for risk assessment purposes, such as protective and control measures or global overviews of AI in a specific region. Both tools are prone to classifying human-related reports as animal-related events. When correctly identified, the detection of zoonotic events in humans is highly relevant from a health perspective. The automatic fine-grained topic classification of news reports still needs improvement to enable discrimination of outbreak declarations from other topics, thus avoiding false alerts and facilitating the triage of sanitary information [21].

In this study, PADI-web was more sensitive than HealthMap, detecting 139 more events than HealthMap. However, the proportion of early detected events compared to the total number of detected events was higher for HealthMap (43% vs. 23%). These differences in captured events may reflect the different web scraping and filtering methods for online news monitoring of the PADI-web and HealthMap. PADI-web is an entirely automatised tool; thus, it captures and filters outbreak-related information without any human intervention. HealthMap is a semi-automatised tool with human moderators that filter news reports that will be shared with users. This may suggest that HealthMap moderators filter and keep only emerging exceptional

AI events (such as primary cases), rather than all possible AI events (primary and secondary cases).

Our study highlights the complementarity of these two EBS tools. This complementarity reflects the different sources accessed through the EBS pipelines. Our results showed, for instance, that PADI-web captured more local sources than HealthMap, while the latter relied more heavily on social platforms such as Twitter. Barboza et al. [10] showed that the EBS tool characteristics such as the type of moderation, sources accessed, diseases, languages, and regions covered significantly influence disease detection performance, and that the system's outbreak detection is synergic (complementary). While the proportion of early detected events in our study may seem modest, it is a significant added value to the EBS regarding the reporting of outbreaks of pathogens with zoonotic and pandemic potential. In addition, both networks were highly reactive, mostly propagating information from primary sources to the aggregator in less than one day. Early detection of public health hazards constitutes a fundamental component of efficient outbreak management [22]. It may be the main determinant in selecting the appropriate response, thus minimising morbidity and mortality caused by an infectious disease [23]. Event-based surveillance should not be considered a replacement for traditional indicator-based surveillance, but rather, complementary to routinely collected public health surveillance data.

While the reporting of AI events by the EBS tools was highly effective, timely, and reactive, a bottleneck may arise at the step of manual analysis of the detected events. The strength of EBS relies heavily on adequate human resources to feed decision-making chains based on detected events. Therefore, in our future work, we will explore how the detected events can be useful for risk assessment and risk mapping.

## Role of the sources

Our results highlight three groups of sources regarding their role in the dissemination of outbreak-related information. EBS tools are aggregators. It is important to note that our results did not reflect ProMED-mail intrinsic performance as an EBS tool, that is, expert network sharing outbreak-related information, but as an intermediate source of HealthMap. Local and national authorities and veterinarians were emitters and were the most important primary sources of events. They produce information that is acknowledged at the local/national level, mostly verified by laboratory tests, and is susceptible to being reported in the media. WOAH, online news, press agencies, social media, and several research organisations combined both abilities by collecting information from a wide range of sources and being highly visible by collector sources in the network (online news, EBS tools). Network performance was driven by the presence of a small number of sources with high individual all-degrees, such as WOAH, Reuters, Xinhua, and several social network platforms. These sources played the role of hubs, not only filtering and disseminating information but also ensuring a connection between different groups in the network [19]. The presence of hubs was not the only feature of network performance, as early detection mostly relied on online news sources with individual low all-degrees. Thus, the early components of EBS networks also relied on their ability to monitor a large number of individually low-performant sources.

National online news plays a major role in early detection by disseminating announcements from local and national veterinary authorities, thus making them detectable by EBS tools. Zhang et al. found out that national newspapers (referred to as "local" newspapers in their methods) provided more specific information about the local Zika virus emergence in Brazil than did international newspapers; similar findings were made for outbreak detection in Nepal [12]. In this study, local sources were more likely to identify a unique event than

international sources, indicating that international sources were more likely to be redundant by publishing multiple reports about the same event [12]. This emphasises the need to target local and national sources available on the web, going beyond sources published in English. The monitoring of multi-lingual sources, integrated into the two EBS tools in our work, is a prerequisite for maximising access to national and local media. The retrieval and analysis of non-English texts have been enhanced and facilitated by the improvement of methods for multi-lingual text processing, such as textual classification [24, 25] and deep-learning-based translation [26]. We believe that efforts to integrate multi-lingual sources will benefit both the Se and timeliness of EBS tools.

Social platforms, mostly used by HealthMap, include generic platforms such as Twitter, but also specialised blogs such as FluTrackers and AvianFluDiary. Specialised blogs are relevant sources for integration into EBS, as they rely on the collection of information from numerous sources, as highlighted by their high median in-degree, previously filtered by domain-specialised moderators. Health blogs were found to cite less sources than online news in a study evaluating H1N1/Swine Flu coverage in the media [27], which is not in line with the highest in-degree found in our study. However, the difference in the number and nature of sources evaluated (eight online news [27]) makes the study hardly comparable. They also translated news from national languages into English, facilitating access to local field information. In addition, owing to their non-official status, online blogs are more prone to communicate events before official notifications. While the classical method of web monitoring is traditionally keyword-oriented (e.g., systematic monitoring of combinations of keywords), source-based monitoring (i.e., systematic monitoring of a specific source) is a costless and easy way to improve existing EBS tools. For instance, retrieving news directly from official government health websites would enhance the geographic representativeness of news aggregators such as Google News [28, 29].

It is important to note that our results were specific to the model disease and study period. For example, the Bulgarian veterinary authority appeared to be an important source because 22 outbreaks were observed in Bulgaria during the study period, including a new incursion of the Highly Pathogenic Avian Influenza (HPAI) H5N8 subtype [30] widely reported by Bulgarian media.

## Re-thinking the role of event-based surveillance in epidemic intelligence

EBS is sometimes opposed to indicator-based surveillance, as it is based on the use of so-called nonofficial sources. In our study, official veterinary authorities (national or local) represented 80% of primary sources, including those involved in early detection. Thus, the monitoring of the PADI-web and HealthMap was mainly characterised by the detection of national or local official events. This detection includes both the dissemination of WOAH-notified outbreaks (late detection) and the dissemination of official events that have not yet been notified (early detection). In the latter case, EBS tools bypass the international notification procedure and its inherent delays. These findings are consistent with the latest and broader definitions of EBS, stating that media sources collected in the context of EBS can be either official (e.g. a Ministry of Health website) or non-official (e.g. newspaper) [31].

Although the extraction of epidemiological information from collected reports has been widely studied, the automatic extraction of cited sources of events from online sources has not yet received attention. However, based on the findings of our study, we believe that this feature would enhance informal surveillance by enabling the characterisation of an event as official at the international, national, or local level, depending on whether the cited source is the WOAH, a national/local veterinary authority, or non-official, if the type of source does not

belong to any of the latest categories. Recent advances in named entity extraction, involving deep learning, combined with a step of normalisation (dictionary or ontology-based), would enable easy identification of the mentioned cited sources. Alerts could be triggered when WOAH is not mentioned. By providing our corpus and databases with open access, we offer the possibility of evaluating and comparing approaches with a high-quality validation dataset.

Both the EBS tools detected several events that could not be found in the EMPRES-i database (S10 Table). These events may have been local AI events that were not communicated at the international level; thus, they did not appear in the EMPRES-i database. They may also correspond to a suspected event that was negated after a negative laboratory test result for the AI virus or to a false alert, as mentioned in a previous study [32]. Thus, our study shows that EBS tools can be a source of relevant outbreak information but should be considered complementary to official sources and interpreted with caution. The identification and characterisation of the sources linked in an EBS are important for prioritising the ones regarding truthfulness and reliability. It may be a way of dealing with fake news, for example, by targeting specialised sources. Our study sets the first list of these sources. By extending our approach to emerging zoonotic infectious diseases, the corpora of reliable news sources may be enriched.

## Conclusion

Current EBS tools use a diverse, but not identical, network of sources; thus, they can be used in parallel by EI practitioners. In addition, both EBS tools should prioritise specialised media sources and access, when existing, to local and national veterinary authorities' webpages, as they released part of the official event before the international notification to the WOAH. Outbreak-related news travels from a primary source to a final aggregator in one day or less, which is important for early warnings and EI. Both PADI-web and HealthMap shared timely outbreak information on AI in domestic and wild birds, thus contributing to the early detection of EI and as complementary sources to traditional surveillance.

A potential future work could be the integration of the results highlighted in this study to improve EBS systems (for instance, by weighting type of sources in EBS platforms). As mentioned in this paper, we can cite multi-lingual aspects to consider for improving the proposed analysis as well as EBS systems. We could evoke the same type of analysis to conduct with other platforms as well, such as ProMED-mail.

## Supporting information

**S1 Table. Definitions used to characterize the types of sources, specialization and geographical focus in PADI-web and HealthMap networks.**
(DOCX)

**S2 Table. Summary of the manual curation of the relevance of PADI-web and HealthMap reports.**
(DOCX)

**S3 Table. The number of events reported to the WOAH and detected by the two EBS tools per week (mean, min, and max) and per region.**
(DOCX)

**S4 Table. Legend of the node's names in the PADI-web network.**
(CSV)

**S5 Table. Legend of the node's names in the HealthMap network.**
(CSV)

**S6 Table. PADI-web network composition.**
(CSV)

**S7 Table. HealthMap network composition.**
(CSV)

**S8 Table.** Proportion of the types of sources according to their role in the (a) PADI-web and (b) HealthMap late detection networks.
(DOCX)

**S9 Table.** Proportion of the types of sources according to their role in the (a) PADI-web and (b) HealthMap early detection networks.
(DOCX)

**S10 Table. Type of primary and secondary sources involved in the detection and transmission of non-official events in PADI-web and HealthMap networks.**
(DOCX)

**S1 File. Statistical comparison of the sensitivity of PADI-web and HealthMap networks.**
(PDF)

**S1 Fig.** PADI-web (A) and HealthMap (B) networks. Sources were grouped by type. The edge colour corresponds to the colour of the incoming source type, thus enabling the visualisation of the direction of information dissemination, that is, orange edges represent incoming edges to an EBS tool.
(TIF)

**S2 Fig. Type of specialization of primary and secondary sources for the detection of early and late events in PADI-web and HealthMap networks.**
(TIF)

## Acknowledgments

We thank the HealthMap project (https://healthmap.org/), which kindly provided us with their data. We acknowledge the reviewer for their constructive comments.

## Author Contributions

**Conceptualization:** Sarah Valentin, Mathieu Roche, Renaud Lancelot, Elena Arsevska.

**Data curation:** Sarah Valentin, Bahdja Boudoua, Kara Sewalk, Elena Arsevska.

**Formal analysis:** Sarah Valentin, Bahdja Boudoua.

**Methodology:** Sarah Valentin, Elena Arsevska.

**Resources:** Mathieu Roche, Renaud Lancelot.

**Supervision:** Mathieu Roche, Renaud Lancelot.

**Validation:** Sarah Valentin.

**Visualization:** Nejat Arınık.

**Writing – original draft:** Sarah Valentin, Bahdja Boudoua, Elena Arsevska.

**Writing – review & editing:** Sarah Valentin, Bahdja Boudoua, Kara Sewalk, Nejat Arınık, Mathieu Roche, Renaud Lancelot, Elena Arsevska.

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
