## [Decision Letter · Decision Letter 0]

11 Oct 2022

PONE-D-22-24102Dissemination of information in event-based surveillance, a case study of Avian InfluenzaPLOS ONE

Dear Dr. Arsevska,

Thank you for submitting your manuscript to PLOS ONE. After careful consideration, we feel that it has merit but does not fully meet PLOS ONE’s publication criteria as it currently stands. Therefore, we invite you to submit a revised version of the manuscript that addresses the points raised during the review process. Please address all the comments and major issues pointed out by the reviewers.

We look forward to receiving your revised manuscript.

Kind regards,

Maira Aguiar, PhD

Academic Editor

PLOS ONE

Journal Requirements:

"This work has been funded by the “Monitoring outbreak events for disease surveillance in a data science context" (MOOD) project from the European Union’s Horizon 2020 research and innovation program under grant agreement No. 874850 (https://mood-h2020.eu/) and is catalogued as MOOD 049."

5. Please upload a new copy of Figure 3 as the detail is not clear. Please follow the link for more information:

https://blogs.plos.org/plos/2019/06/looking-good-tips-for-creating-your-plos-figures-graphics/

https://blogs.plos.org/plos/2019/06/looking-good-tips-for-creating-your-plos-figures-graphics/

Reviewers' comments:

Reviewer's Responses to Questions

**Comments to the Author**

1. Is the manuscript technically sound, and do the data support the conclusions?

Reviewer #1: Yes

Reviewer #2: Partly

2. Has the statistical analysis been performed appropriately and rigorously? 

Reviewer #1: Yes

Reviewer #2: No

3. Have the authors made all data underlying the findings in their manuscript fully available?

Reviewer #1: Yes

Reviewer #2: Yes

4. Is the manuscript presented in an intelligible fashion and written in standard English?

Reviewer #1: Yes

Reviewer #2: Yes

5. Review Comments to the Author

Reviewer #1: Valentin et al describes how event-based surveillance are propagated by different networks. In the XXI century, where social media are highly important for dissemination news, it is important to check if disease outbreaks, especially those caused by zoonotic viruses, are accurate. Therefore, I have some minor reviews I hope it will help other readers before its publication.

Line 35: Please write what WOAH means

Line 165: there’s a N staring the sentence (also in lines 276 and 278 that are starting with numbers). Please check

Within the results section, what do authors mean by unique events in Table 1?

Figure 3 is impossible to read. Could the authors improve the image quality?

Reviewer #2: 

Introduction –

First paragraph : The manuscript refers to communication in health surveillance and how it can be expanded in the case of avian influenza. Which bibliographic reference of the world health organization that guides or suggests the use of the dissemination of information on health-related events?

What context do these Padi Web and Health Map applications work in? The first paragraphs do not mention health surveillance and its emergencies where these programs/applications can be useful.

Second paragraph: it is not clear and explanatory all the advantages of using healthy maps descriptors. It must be in simple and clear computational language, after all, the target audience is not only the scientific community, but health workers

Seventh paragraph, last line: What is your source of comparison in relation to the healthy map data? what is the assumption or hypothesis that it can be more useful ?

Regarding the questions of this work:

1. What are the sources involved in the reporting of outbreak-related information on the

web?- This would not be a question but a methodology to evaluate.

2. What is the role of the different sources regarding the spread of outbreak-related

information on the web and what are their characteristics in terms of type, specialization

and geographical scope? OK

3. How complementary are the different EBS tools in terms of monitored sources and

reported outbreak-related information?—Is it compared to which data?

Methodology

Event detection

First paragraph: We chose a one-year 131 study period (July 2018 - June 2019) to capture the space-time epidemiological characteristics of the AI outbreaks around the world. From which agencies?What sources?

Define about Empres-i - How it collects health data from official sources?

Second paragraph line 145, define what this acronym WOAH means. From this description you can mention only the acronym but not have defined yourself previously

Network construction

First paragraph

“We assumed that an information pathway could be deducted from the sources cited in a news content. In na information pathway, the first node is called the primary source (i.e. the earliest emitter source), the last node is called the final source (i.e. the final aggregator, PADI-web orHealthMap) and the remaining nodes, if any, are called secondary sources.”

Comment: It is necessary to modify this definition because primary data in public health and epidemiology are those obtained directly in the territory to be sampled regarding a certain disease data. A secondary data are obtained through the country's information systems.

No reference to the global surveillance system by a specific WHO program was cited or used (https://www.who.int/initiatives/global-influenza-surveillance-and-response-system and https://www.who. int/health-topics/influenza-avian-and-other-zoonotic#tab=tab_1) Why?

Methodology

Official sources on animal and human surveillance should not be test sources for the network as they are the gold standard for comparing sources of risk communication.

Qualitative nodes analysis:

Reformulate or change the terms referring to primary and secondary data that cannot refer to the EBS tools technique because they are intrinsically used terms. The terms used must be from epidemiology.

How sensitive/specific is the PADI web and Health Map data compared to the gold standard of data? Where are the statistical analyzes showing this fact?

As for the geographic scope, it was not clear in the text to the national scope that the data refer

The data should cover the following variables: total number and frequencies of avian influenza events; mean, maximum and minimum value of the number of events monitored per epidemiological week; source and means of event notification; frequency of events monitored by region of occurrence and spatial distribution of events according to reference municipality; opportunity to notification; Closing opportunity (time interval between the date from the notification to the National Surveillance until the end of its monitoring) classification of the group of events according to means of transmission and risk classification after evaluation of the events

6. PLOS authors have the option to publish the peer review history of their article (what does this mean?). If published, this will include your full peer review and any attached files.

Reviewer #1: No

Reviewer #2: No

---

## [Author Response · Author response to Decision Letter 0]

5 Feb 2023

November 16, 2022

Subject: Response to the review of manuscript number PONE-D-22-24102

Dear PlosOne Chief Editor and Reviewers,

We acknowledge your comments on our manuscript ”Dissemination of information in event-based surveillance,

a case study of Avian Influenza”. We addressed your constructive reviews by modifying our

manuscript (using track changes) and answering the reviewers’ questions here-below.

Best regards,

The authors

General comments from the editor

If applicable, we recommend that you deposit your laboratory protocols in protocols.io to enhance the

reproducibility of your results. Protocols.io assigns your protocol its own identifier (DOI) so that it can

be cited independently in the future. For instructions see: https://journals.plos.org/plosone/s/

submission-guidelines#loc-laboratory-protocols.

1. Please ensure that your manuscript meets PLOS ONE’s style requirements, including those for file

naming. The PLOS ONE style templates can be found at :

- https://journals.plos.org/plosone/s/file?id=wjVg/PLOSOne_formatting_sample_main_body.

pdf, and

- https://journals.plos.org/plosone/s/file?id=ba62/PLOSOne_formatting_sample_title_authors_

affiliations.pdf

- Author affiliations formatting. We have added the appropriate pilcrow symbol for the equal contributors

of the work. We have set the appropriate format for the corresponding author. We have fixed the

affiliations, by removing postcodes and removing abbreviations of Departments and listing all institutions

in full. Please check page 1 of the manuscript.

- Manuscript body formatting. We have adjusted level 1 heading for all major sections. File formats for

figures were corrected, now they are in .tiff format and passed via the PACE tool suggested by PlosOne.

2. We note that the grant information you provided in the ‘Funding Information’ and ‘Financial Disclosure’

sections do not match. When you resubmit, please ensure that you provide the correct grant

numbers for the awards you received for your study in the ‘Funding Information’ section.

- Done. Funding from Acknowledgments section has been removed and moved into the ‘Funding Information’

and ‘Financial Disclosure’ sections. Please see the new Acknowledgments section in line 546.

3. Thank you for stating the following in the Acknowledgments Section of your manuscript: ”This work

has been funded by the “Monitoring outbreak events for disease surveillance in a data science context”

(MOOD) project from the European Union’s Horizon 2020 research and innovation program under grant

agreement No. 874850 (https://mood-h2020.eu/) and is catalogued as MOOD 049.”

We note that you have provided funding information that is not currently declared in your Funding

Statement. However, funding information should not appear in the Acknowledgments section or other

areas of your manuscript. We will only publish funding information present in the Funding Statement

section of the online submission form.

Please remove any funding-related text from the manuscript and let us know how you would like to

update your Funding Statement. Currently, your Funding Statement reads as follows: ”The funders

had no role in study design, data collection and analysis, decision to publish, or preparation of the

manuscript.” Please include your amended statements within your cover letter; we will change the online

submission form on your behalf.

- Done. Funding from Acknowledgments section has been removed and moved into the ‘Funding Information’

and ‘Financial Disclosure’ sections.

- Please continue to use the current Funding Statement: ”The funders had no role in study design, data

collection and analysis, decision to publish, or preparation of the manuscript.”

4. In your Data Availability statement, you have not specified where the minimal data set underlying

the results described in your manuscript can be found. PLOS defines a study’s minimal data set as

the underlying data used to reach the conclusions drawn in the manuscript and any additional data

required to replicate the reported study findings in their entirety. All PLOS journals require that the

minimal data set be made fully available. For more information about our data policy, please see http:

//journals.plos.org/plosone/s/data-availability. Upon re-submitting your revised manuscript,

please upload your study’s minimal underlying data set as either Supporting Information files or to a

stable, public repository and include the relevant URLs, DOIs, or accession numbers within your revised

cover letter. For a list of acceptable repositories, please see http://journals.plos.org/plosone/s/

data-availability#loc-recommended-repositories. Any potentially identifying patient information

must be fully anonymized.

- We created a Zenodo repository (https://doi.org/10.5281/zenodo.7324144) containing the entire

dataset to reproduce the results. We provided the link in the manuscript, section Data reporting, line

549.

- We also shared the script for our results presented in the manuscript in a public GitHub repository

(https://github.com/SarahVal/EBS-network). We provided the link in the manuscript, section Statistical

reporting, line 552.

- Our dataset does not contain patient information.

Important: If there are ethical or legal restrictions to sharing your data publicly, please explain these

restrictions in detail. Please see our guidelines for more information on what we consider unacceptable restrictions

to publicly sharing data: http://journals.plos.org/plosone/s/data-availability#locunacceptable-

data-access-restrictions. Note that it is not acceptable for the authors to be the sole

named individuals responsible for ensuring data access. We will update your Data Availability statement

to reflect the information you provide in your cover letter.

- There are no legal and ethical restrictions for sharing our dataset publicly. Please check the description

of our dataset at: https://doi.org/10.5281/zenodo.6908000

5. Please upload a new copy of Figure 3 as the detail is not clear. Please follow the link for more information:

https://blogs.plos.org/plos/2019/06/looking-good-tips-for-creating-your-plos-figuresgraphics/

- All figures have passed though the PACE web-based imaging review tool. We provide you with new

figure publication graphics in a .tiff format, uploaded separately. For clarity, we have moved Figure 3

into Supp material.

Comments from reviewer 1

Line 35: Please write what WOAH means.

- Done, we defined World Organisation for Animal Health (WOAH, founded as OIE), line 159. We

further checked for all other acronyms and their first mention full description.

Line 165: there’s a N staring the sentence (also in lines 276 and 278 that are starting with numbers).

Please check

- Removed in line 165, it was a typing error. However, we did not find typos for numbers for lines 276 &

278.

Within the results section, what do authors mean by unique events in Table 1?

- A unique event, non-overlapping event, as initially defined in our manuscript, was an event detected by

either of the event-based surveillance (EBS) tools, PADI-web or HealthMap. More precisely, a unique

event was an event event detected by PADI-web (or by HealthMap, respectively) and not detected by

HealthMap (or by PADI-web, respectively). To avoid confusion, we replace the term ”unique” by ”nonoverlapping”.

Non-overlapping events enable us to analyse the overlap (and, thus, the complementary)

between HealthMap and PADI-web. We provide an improved description of the term ”unique event” in

the manuscript in the section Material and methods, section Event detection line 166 and in the Results,

section Event detection lines 266-271.

Figure 3 is impossible to read. Could the authors improve the image quality?

- All figures have passed though the PACE web-based imaging review tool. We provide you with new

figure publication graphics in a .tiff format, uploaded separately. For clarity, we have moved Figure 3

into Supp material.

Comments from reviewer 2

Introduction

First paragraph: The manuscript refers to communication in health surveillance and how it can be expanded

in the case of avian influenza. Which bibliographic reference of the world health organization that

guides or suggests the use of the dissemination of information on health-related events?

- We added references to the Epidemic Intelligence paradigm, which promotes the use of non-official

sources to follow the dissemination of information on health-related events and complement indicatorbased

surveillance. We have in detail reworked the introduction, please check pages 3 and 4.

What context do these Padi-web and HealthMap applications work in? The first paragraphs do not

mention health surveillance and its emergencies where these programs/applications can be useful.

- PADI-web and HealthMap facilitate the collection, analysis and dissemination of event-based surveillance

data on infectious diseases and associated health issues, in the context of epidemic intelligence.

Several studies have assessed their use and performances in different epidemiological contexts including

new and enzootic, epizootic and zoonotic infectious diseases. We provide example and new references in

the manuscript. We have in detail reworked the introduction, please check pages 3 and 4.

Second paragraph: it is not clear and explanatory all the advantages of using healthy maps descriptors. It

must be in simple and clear computational language, after all, the target audience is not only the scientific

community, but health workers.

We specified the audience and simplified the description of both tools in the manuscript. We have in

detail reworked the introduction, please check pages 3 and 4.

-Seventh paragraph, last line: What is your source of comparison in relation to the healthy map data?

what is the assumption or hypothesis that it can be more useful ?

- In the seventh paragraph, we refer to a former study that evaluated the role of the sources detected

by HealthMap regarding the detection of outbreaks, at a national scale (Nepal). The gold standard

database with which the authors compared HealthMap was the official country outbreak notifications.

We motivate our study as an extension of this work, by providing two significant enhancements: (1) we

enlarge this work on a global scale and (2) we do not solely rely on the sources directly detected by the

EBS tools, but we trace back the origin of the outbreak information. We have in detail reworked the

introduction, please check pages 3 and 4.

Regarding the questions of this work

1. What are the sources involved in the reporting of outbreak-related information on the web?- This would

not be a question but a methodology to evaluate.

- Every EBS media monitoring tool in use today has its own methodology for detection of sources on the

web, collection, filtering of news and extraction of relevant information from the unstructured text from

the news. The sources detected by an EBS tool result from (1) the choice of targeting a specific source

(e.g. HealthMap collect Pro-MED alerts) and (2) its methodological choices (e.g. keywords to capture

the news, languages for the keywords, Google news regions to monitor, etc.). In the last case, the specific

online news that will be captured cannot be know a priori. In our work, we do not solely evaluate the

sources directly detected by the EBS tools, but, we also trace back and characterise the initial sources

first emitting the disease outbreak information (referred to as primary sources in our manuscript) and

the intermediate ones, based on the manual evaluation of all sources cited in each news, which was a

fastidious work of data collection and curation for the co-authors. We provide a clarification on this

objective in the introduction.

3. How complementary are the different EBS tools in terms of monitored sources and reported outbreakrelated

information?—Is it compared to which data?

We address this question in two steps. First, we calculate the proportion of overlapping events (events

that were detected by both PADI-web and HealthMap), We show that almost half of the detected events

were non-overlapping events. Second, we show that the two tools do not monitor the same sources (i.e.

PADI-web retrieved a largest number of online news sources, while HealthMap retrieved content from

more social platforms than PADI-web). Please check, the Event detection section in Methods, lines

151-167 and in Results, lines 251-271.

Methodology

Event detection

First paragraph: We chose a one-year 131 study period (July 2018 - June 2019) to capture the spacetime

epidemiological characteristics of the AI outbreaks around the world.–¿ From which agencies?What

sources?

The official data source is described further in our manuscript (Empres-i). Here, we meant that we

wanted to embrace a time period enabling us to capture different epizootic events worldwide, to be able

to compare the EBS tools and evaluate the network of sources based on a large number of AI outbreaks.

Please check lines 151-165.

- We provide a new sentence in the Methods section: ”We chose a one-year study period (July 2018 -

June 2019) to capture larger scale AI outbreak patterns around the world.” Please check lines 128-135.

Define about Empres-i - How it collects health data from official sources?

- We provide a more clear description of the EMPRES-i database, its purpose and its sources. Please

check the Event detection of the Materials and methods section, lines 151-165..

Second paragraph line 145, define what this acronym WOAH means. From this description you can

mention only the acronym but not have defined yourself previously

- Done, we provide the full name of the World Organisation for Animal Health (WOAH, ex-OIE). Please

check line 159.

Network construction

First paragraph “We assumed that an information pathway could be deducted from the sources cited

in a news content. In an information pathway, the first node is called the primary source (i.e. the

earliest emitter source), the last node is called the final source (i.e. the final aggregator, PADI-web or

HealthMap) and the remaining nodes, if any, are called secondary sources.” Comment: It is necessary to

modify this definition because primary data in public health and epidemiology are those obtained directly

in the territory to be sampled regarding a certain disease data. A secondary data are obtained through

the country’s information systems.

Epidemic intelligence (EI) encompasses all activities related to early identification of potential health

hazards, their verification, assessment and investigation in order to recommend public health control measures.

EI integrates both an indicator-based and an event-based component. ‘Indicator-based component’

refers to structured data collected through routine surveillance systems, corresponding to the definitions

provided by the reviewer. ‘Event-based component’, the context of our study, refers to unstructured data

gathered from sources of intelligence of any nature (e.g. media, laboratory, channels of communications,

etc.,see https://www.eurosurveillance.org/content/10.2807/esm.11.12.00665-en). As noted by

the reviewer, the primary sources in terms of diagnosis is usually a laboratory, even in EBS, especially

when studying a well-known disease subject to notification as avian influenza. However, this is not true

when the detected disease is not yet diagnosed and when solely information about unusual symptoms are

communicated. This component of EBS, which is closed to the syndromic surveillance, is an essential

component of early detection. In this study, we defined primary sources in EBS paradigm as the earliest

cited source of each path, which is not necessarily the primary source in terms of diagnosis, but rather

in terms of communication. Thus, it can include official sources typically involved in IBS (laboratory,

country’s official authorities), as well as informal sources (a person, an company, etc.). We have reworked

the introduction, please check pages 3 and 4.

No reference to the global surveillance system by a specific WHO program was cited or used (https: //

www. who. int/ initiatives/ global-influenza-surveillance-and-response-system and https:

// www. who. int/ health-topics/ influenza-avian-and-other-zoonotic ) Why?

Our study lies in the context of event-based surveillance in the animal health domain. We did not

described World Health Organization surveillance programs as they mainly focus on zoonotic events

from a public health perspective, in the indicator-based paradigm. Besides, our objective was to describe

the EBS systems.

Official sources on animal and human surveillance should not be test sources for the network as they are

the gold standard for comparing sources of risk communication. In this study, official sources on animal

and human surveillance are not tested by themselves. They appeared in the network because they were

cited by non-official sources monitored bu the EBS tools. For instance, if an online news sources stated

”According to the WHOA, an outbreak of avian influenza was detected yesterday in country X”, WHOA

was the emitter (primary) source of our network.

Qualitative nodes analysis: Reformulate or change the terms referring to primary and secondary data

that cannot refer to the EBS tools technique because they are intrinsically used terms. The terms used

must be from epidemiology.

To our knowledge, this work is the first attempt to describe the dissemination of information between

sources cited in online news in the context of health surveillance, and no specific terms where proposed to

refer to such sources in the epidemiological context. Thus, we proposed the terms primary and secondary

as they are explicit for the reader and reflect the temporal diffusion of the events.

How sensitive/specific is the PADI web and Health Map data compared to the gold standard of data?

Where are the statistical analyzes showing this fact?

-We calculated the sensitivity of HealthMap and PADI-web, following the definition provided in section

Methods. The specificity of event-based surveillance tools cannot be calculated, as it is impossible to

assess the status of non-official events they detect; there may be false positive events, as well as true

positive events not reported to the gold standard databases (WOAH and EMPRES-i). We did not

provide any further statistical tests as the purpose of our study is not to evaluate the influence of factors

in the sensitivity of the tools. Please check the apprach and the results in lines 168-181 and 276-278.

As for the geographic scope, it was not clear in the text to the national scope that the data refer. The

data should cover the following variables: total number and frequencies of avian influenza events; mean,

maximum and minimum value of the number of events monitored per epidemiological week; source and

means of event notification; frequency of events monitored by region of occurrence and spatial distribution

of events according to reference municipality; opportunity to notification; Closing opportunity (time

interval between the date from the notification to the National Surveillance until the end of its monitoring)

classification of the group of events according to means of transmission and risk classification after

evaluation of the events

For the data from EBS tools, we did not chose any national scope a priori: our data selection was solely

based on the studied disease (avian influenza) and host (animals) worldwide. To clarify, we added a

table summarizing the total number and frequencies of avian influenza events; mean, maximum and

minimum value of the number of events monitored per week; and the source of the event notification as

Supplementary material.

---

## [Decision Letter · Decision Letter 1]

10 Apr 2023

PONE-D-22-24102R1Dissemination of information in event-based surveillance, a case study of Avian InfluenzaPLOS ONE

Dear Dr. Arsevska,

Thank you for submitting your manuscript to PLOS ONE. After careful consideration, we feel that it has merit but does not fully meet PLOS ONE’s publication criteria as it currently stands. Therefore, we invite you to submit a revised version of the manuscript that addresses the points raised during the review process.

Please consider including the reviewers comments before submitting your manuscript, especially regarding the statistical analysis performed in the study.

We look forward to receiving your revised manuscript.

Kind regards,

Maira Aguiar, PhD

Academic Editor

PLOS ONE

Journal Requirements:

Reviewers' comments:

Reviewer's Responses to Questions

**Comments to the Author**

1. If the authors have adequately addressed your comments raised in a previous round of review and you feel that this manuscript is now acceptable for publication, you may indicate that here to bypass the “Comments to the Author” section, enter your conflict of interest statement in the “Confidential to Editor” section, and submit your "Accept" recommendation.

Reviewer #1: All comments have been addressed

Reviewer #2: All comments have been addressed

2. Is the manuscript technically sound, and do the data support the conclusions?

Reviewer #1: Yes

Reviewer #2: Partly

3. Has the statistical analysis been performed appropriately and rigorously? 

Reviewer #1: Yes

Reviewer #2: No

4. Have the authors made all data underlying the findings in their manuscript fully available?

Reviewer #1: Yes

Reviewer #2: Yes

5. Is the manuscript presented in an intelligible fashion and written in standard English?

Reviewer #1: Yes

Reviewer #2: Yes

6. Review Comments to the Author

Reviewer #1: Authors have improved the manuscript, which is now suitable for publication. Figures, that were in bad resolution in the former revision, were also improved.

Reviewer #2: The previously suggested corrections were accepted and now the text is much more didactic and transparent for reading by authors from other areas of knowledge.The coments are in dialog boxes distributed in the body of the text of the manuscript.

7. PLOS authors have the option to publish the peer review history of their article (what does this mean?). If published, this will include your full peer review and any attached files.

Reviewer #1: No

Reviewer #2: No

---

## [Author Response · Author response to Decision Letter 1]

14 Apr 2023

Subject: Response to the review of manuscript number PONE-D-22-24102

Dear PlosOne Chief Editor and Reviewers,

We acknowledge your comments on our manuscript ”Dissemination of information in event-based surveillance,

a case study of Avian Influenza”. We addressed the remaining issues in our manuscript (using track

changes) and provide a response to the reviewers’ questions here-below. Please note that we detected an

artefact in Figure 4 which was re-exported.

Best regards,

The authors

Comments from reviewer 1

Whats the standardizing of the ”relevant news and irrelevant news”?

We did not standardize relevant and irrelevant news. We categorized them manually by relevance. As

described in the manuscript, relevant news was describing a disease outbreak and irrelevant news was

talking about another topic, irrelevant to a disease outbreak. Thus we did not need to standardise the

news, we just categorized them.

We evaluated the Se? What is this? There is no acronym that explains it?

We referred to the sensitivity (Se) before defining the acronym. We fixed the issue. In the event-based

surveillance context, sensitivity corresponds to the number of official events detected by the EBS system.

How can the padi web software be considered more sensitive than Health map when there is no kappa

value, or fisher test or other robust statistics used in the analyses?

We added a statistical test (McNemar’s test on paired nominal data for, i.e., to compare the proportions

for two correlated dichotomous variables) to evaluate the significance of the difference between the sensitivity

(Se), at the standard significance level of 5% (the analysis is provided as supplementary material,

S1 File.pdf). We provided precision in the discussion, in the phrase on the sensitivity between the two

systems.

---

## [Editor Report · Decision Letter 2]

20 Apr 2023

Dissemination of information in event-based surveillance, a case study of Avian Influenza

PONE-D-22-24102R2

Dear Dr. Arsevska,

We’re pleased to inform you that your manuscript has been judged scientifically suitable for publication and will be formally accepted for publication once it meets all outstanding technical requirements.

Kind regards,

Maira Aguiar, PhD

Academic Editor

PLOS ONE
---

## [Editor Report · Acceptance letter]

10 May 2023

PONE-D-22-24102R2 

Dissemination of information in event-based surveillance, a case study of Avian Influenza 

Dear Dr. Arsevska:

I'm pleased to inform you that your manuscript has been deemed suitable for publication in PLOS ONE. Congratulations! Your manuscript is now with our production department. 

Kind regards, 

on behalf of

Dr. Maira Aguiar 

Academic Editor

PLOS ONE